# A mitogenomic timetree for Darwin's enigmatic South American mammal *Macrauchenia patachonica*

Michael Westbury[1], Sina Baleka[1], Axel Barlow[1], Stefanie Hartmann[1], Johanna L.A. Paijmans[1], Alejandro Kramarz[2], Analía M. Forasiepi[3], Mariano Bond[4], Javier N. Gelfo[4], Marcelo A. Reguero[4], Patricio López-Mendoza[5], Matias Taglioretti[6], Fernando Scaglia[6], Andrés Rinderknecht[7], Washington Jones[7], Francisco Mena[8], Guillaume Billet[9], Christian de Muizon[9], José Luis Aguilar[10], Ross D.E. MacPhee[11] & Michael Hofreiter[1]

The unusual mix of morphological traits displayed by extinct South American native ungulates (SANUs) confounded both Charles Darwin, who first discovered them, and Richard Owen, who tried to resolve their relationships. Here we report an almost complete mitochondrial genome for the litoptern *Macrauchenia*. Our dated phylogenetic tree places *Macrauchenia* as sister to Perissodactyla, but close to the radiation of major lineages within Laurasiatheria. This position is consistent with a divergence estimate of ∼66 Ma (95% credibility interval, 56.64–77.83 Ma) obtained for the split between *Macrauchenia* and other Panperissodactyla. Combined with their morphological distinctiveness, this evidence supports the positioning of Litopterna (possibly in company with other SANU groups) as a separate order within Laurasiatheria. We also show that, when using strict criteria, extinct taxa marked by deep divergence times and a lack of close living relatives may still be amenable to palaeogenomic analysis through iterative mapping against more distant relatives.

[1] University of Potsdam, Institute of Biochemistry and Biology, Karl-Liebknecht-Str. 24-25, 14476 Potsdam, Germany. [2] CONICET and Sección Paleontología de Vertebrados, Museo Argentino de Ciencias Naturales 'Bernardino Rivadavia', Avenida Angel Gallardo 470, Buenos Aires C1405DJR, Argentina. [3] IANIGLA, CCT-CONICET Mendoza, Av. Ruiz Leal s/nº, Parque General San Martín, Mendoza 5500, Argentina. [4] CONICET and División Paleontología Vertebrados, Museo de La Plata, Paseo del Bosque s/nº, La Plata B1900FWA, Facultad de Ciencias Naturales y Museo, UNLP, Argentina. [5] ARQMAR, Center for Maritime Archeology Research of the South Eastern Pacific, Cochrane 401, Valparaiso, Chile. [6] Museo Municipal de Ciencias Naturales 'Lorenzo Scaglia', Plaza España s/nº, Mar del Plata 7600, Argentina. [7] Museo Nacional de Historia Natural, 25 de Mayo 582, CC 399, Montevideo 11000, Uruguay. [8] Centro de Investigación en Ecosistemas de la Patagonia, Simpson 471, Coyhaique, Chile. [9] Muséum national d'Histoire naturelle, Département Origines et Évolution, CR2P (CNRS, MNHN, UPMC, Sorbonne-Université), 8, rue Buffon, 75005 Paris, France. [10] Museo Paleontológico de San Pedro 'Fray Manuel de Torres', Pellegrini 145, San Pedro 2930, Argentina. [11] Department of Mammalogy, American Museum of Natural History, 200 Central Park West, New York, New York 10024-5192, USA. Correspondence and requests for materials should be addressed to R.D.E.M. (email: macphee@amnh.org) or to M.H. (email: michael.hofreiter@uni-potsdam.de).

It is now well accepted that ancient DNA (aDNA) is a valuable tool for uncovering phylogenetic relationships of extinct animals[1]. However, to obtain correct DNA sequences from ancient remains, it is usual practice to utilize sequences from a close extant relative to produce primer sequences for PCR[1,2], baits for hybridization capture[3,4] or reference frameworks for mapping shotgun data[5]. In theory, reconstructing an ancient genome *de novo* can be undertaken without relying on a close relative's DNA for guidance, but due to contaminant DNA and low average fragment lengths, *de novo* assembly is generally considered not computationally feasible[6,7]. These difficulties are compounded when targeted extinct species lived in tropical or subtropical regions, where aDNA preservation is characteristically poor[7,8]-as in the case of the enigmatic South American mammal *Macrauchenia patachonica*.

*Macrauchenia patachonica* was among the last of the Litopterna, an endemic order whose fossil record extends from the Paleocene to the end of the Pleistocene and includes some 50 described genera. Over the past 180 years, remains of *Macrauchenia* and its close allies have been found in Quaternary deposits in various parts of the continent, first and most notably by Charles Darwin in 1834, near Puerto San Julián in southern Patagonia (Supplementary Note 1). However, neither Darwin nor Richard Owen, who described the species in 1838 (ref. 9), were able to place this taxon securely among placentals[10]. Owen, who had only a few limb bones and vertebrae to work with, originally described *Macrauchenia* as a form 'transitional' between camelids and other ruminant artiodactyls and so-called Pachydermata, a polyphyletic miscellany that included elephants, horses, hippos and hyraxes. Owen's analysis indicated that *Macrauchenia* was, at least in terms of grade, an ungulate of some sort, but it was otherwise inconclusive. Similar uncertainties have marked all subsequent morphology-based efforts to ascertain the affinities not only of Litopterna but also other SANU orders[11–14] (Supplementary Note 1).

The central problem in SANU systematics has long been how to evaluate the remarkable level of similarity individual orders display to various non-SANU taxa from other parts of the world. Unsurprisingly, different studies have reached very different conclusions. One such study[14], utilizing a large set of morphological characters, found that Litopterna belonged within Pan-Euungulata, but another SANU order, Notoungulata, grouped with Afrotheria, indicating that SANUs were not monophyletic. By contrast, utilizing protein (collagen) sequence information, two recently published molecular studies[8,15] found that litopterns as well as notoungulates formed a monophyletic unit that shared more recent common ancestry with Perissodactyla than with any other extant placental group[16] (justifying recognition of the new unranked taxon Panperissodactyla[8]).

Although the collagen (I) evidence for the position of litopterns and notoungulates is informative, given the historical instability of SANU systematics, it is important to corroborate the proteomic results with additional, preferably molecular sources of evidence. However, to date, attempts to use standard aDNA methodologies to collect genetic material from specimens from low-latitude localities have been largely unsuccessful[8]. A promising new approach is shotgun sequencing applied with iterative mapping, which functions in a way similar to reference-assisted *de novo* assembly[17] in bypassing the need for a close relative as reference. Initially, reads are mapped to a specified bait reference sequence. As analysis proceeds, a consensus of mapped reads becomes the new reference for each following iteration until no new reads are found to map. Using this approach, we report the successful collection of mitogenomic data from a South American native ungulate. Our phylogenetic analyses place *Macrauchenia* as a sister taxon to all living Perissodactyla, with the origin of Panperissodactlya at ∼66 Ma, successfully demonstrating that even taxa marked by deep divergence times with no close living relatives are amenable to palaeogenomic analysis.

## Results

**Sample screening.** We extracted DNA from 6 *Macrauchenia* and 11 *Toxodon* bone samples obtained from various sites across South America (Fig. 1, Supplementary Table 1), using a DNA extraction method specifically developed for recovering short fragments typical of aDNA[18]. We converted the resulting extracts into Illumina libraries applying a single-strand library building approach, also specifically developed for aDNA[19] and carried out low-level sequencing (ranging from 2 to 20 million raw reads) to investigate endogenous DNA content. Only a 2nd phalanx, from Baño Nuevo-1 Cave (Coyhaique, Chile, Supplementary Fig. 1) and here coded MAC002, yielded a high number of reads mapping to the horse and rhinoceros nuclear genomes (1.9 and 2.8%, respectively, as opposed to <1% for all others; Supplementary Table 2). This result led us to conduct further shotgun sequencing of this individual for a total of ∼69 million, paired-end 75 bp reads and 43 million after quality controls (Methods section).

**Validation of the iterative mapping approach using MITObim.** The lack of a suitable reference mitochondrial sequence necessitated an iterative mapping approach. We used the iterative mapping software package MITObim[17], which has been used successfully for mitochondrial reconstructions using modern DNA. Reference sequences for the following four species

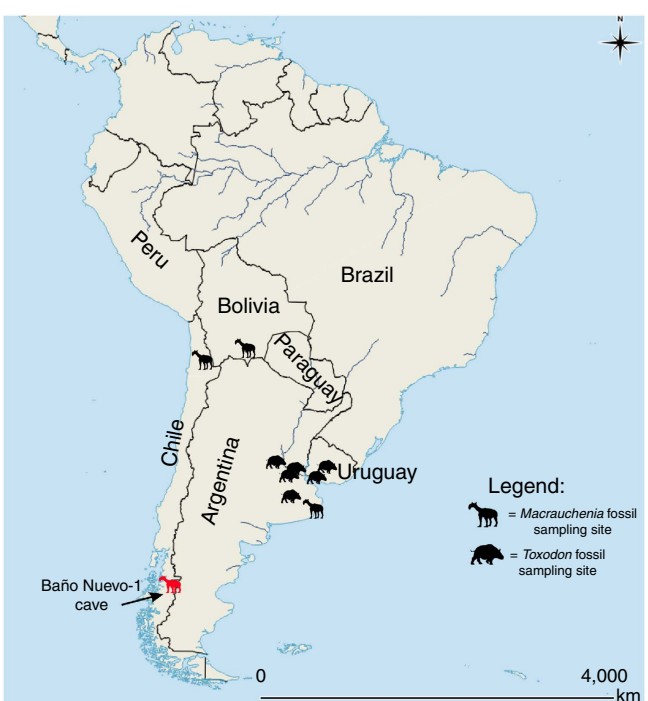

**Figure 1 | Map of sites yielding specimens of *Toxodon* and *Macrauchenia*.** MAC002 *Macrauchenia*, the sample from which mitogenomic data were successfully collected, came from a metapodial recovered at the locality Baño Nuevo-1 Cave (in red). For locality context, see Supplementary Note 1 and Supplementary Fig. 8. Map generated using QGIS 2.8 (QGIS Development Team, 2016. QGIS Geographic Information System. Open Source Geospatial Foundation Project. http://www.qgis.org/).

(with GenBank accession numbers) were selected[17]: guanaco (*Lama guanicoe*, NC_011822.1), rhinoceros (*Ceratotherium simum*, Y07726.1), horse (*Equus caballus*, HQ439492.1) and tapir (*Tapirus indicus*, KJ417810.1). MITObim reconstructions using the default MITObim mismatch value (15%) and consensus calling method resulted in complete mitochondrial sequences being recovered for each bait reference used, but many discrepancies were evident among the consensus sequences (Supplementary Table 3). On visual inspection of the assembly, random read mapping was clearly visible, because large differences between reads mapping to the same region of the bait sequence could be seen. This led us to perform a software validation using Pleistocene cave hyena (*Crocuta crocuta spelaea*) DNA sequences. We compared consensus sequences produced using iterative mapping to distant references with one produced using direct mapping to a cave hyena mitochondrial genome. When using 80% of the average read coverage as the minimum coverage threshold for the consensus sequence base calling, regardless of the mismatch value or reference sequence we tried, the sequence produced with MITObim matched perfectly with the presumably correct consensus sequence produced by direct mapping to the reference cave hyena mitochondrial genome (NC_020670.1; Supplementary Table 4). This result led us to conclude that MITObim was a suitable platform for reconstructing the mitochondrial genome of *Macrauchenia* if appropriately stringent mismatch values and consensus calling parameters were implemented.

**Mitochondrial genome reconstruction of MAC002.** When implementing the 80% average coverage minimum cutoff value as predicted using the cave hyena (Supplementary Table 4), the initial strict value of 0% mismatch between reference and mapping reads produced four consensus sequences displaying exact identity to each other regardless of whether the guanaco, rhinoceros, horse or tapir bait reference was used. As using 0% mismatch value only recovered between ∼10 and 25% of the complete mitochondrial genome (Supplementary Table 5), we relaxed the mismatch value in 1% increments to recover more of the genome.

At a mismatch value of 7%, we noted that a number of sites could not be called unambiguously since discrepancies arose between consensus sequences obtained using different references.

We therefore considered 6% as the upper threshold mismatch value as these disagreements may have arisen due to random read mappings.

Regardless of the implemented mismatch value, when using mismatch values from 0 to 6%, all mappings resulted in alignments of similar average depth (∼40×) and identical sequences, the only difference being in the varying amounts of mitogenome coverage (Supplementary Table 5). The relationship between assembly completeness and mismatch value was, however, not entirely predictable, with some regions of the mitochondrion being covered in assemblies with low mismatch values that were not recovered at higher mismatch values (Supplementary Table 5). The guanaco (artiodactyl) reference produced sequences compatible with those produced using the perissodactyl references, ruling out the possibility of ascertainment biases based on phylogenetic relatedness. In total, we recovered 13,269 basepairs (79.1%) of the *Macrauchenia* mitogenome. Most remaining sites were also covered, but due to our strict consensus calling parameters (Methods section) involving a minimum coverage threshold of 80% of the average, many of these were considered as missing data. Stretches of missing data predominantly occur in the *cytb*, *cox2* and *nad6* genes (Supplementary Fig. 1).

**Macrauchenia mitochondrial sequence validation.** MITOS (ref. 20) automated annotation confirmed the presence of most transfer RNA (tRNA) sequences, apart from glutamic acid and serine, all protein coding genes and both ribosomal RNAs (Supplementary Fig. 1). Amino acid translations of manually predicted protein coding genes showed no indication of premature stop codons. We also conducted an analysis of pairwise sequence identity between our reconstructed *Macrauchenia* sequence and all reference sequences used for mapping as well as the human mitochondrial DNA sequence as outgroup. This analysis showed all regions of the reconstructed *Macrauchenia* mitogenome sequence as approximately equidistant to all reference sequences, with the human mitogenome having a consistently lower pairwise identity throughout (Fig. 2). No regions showed a large increase in pairwise identity to the human sequence, indicating that no regions were constructed from human contaminant DNA. Mapdamage[21] analysis of the mapped reads showed characteristic patterns of DNA damage and short

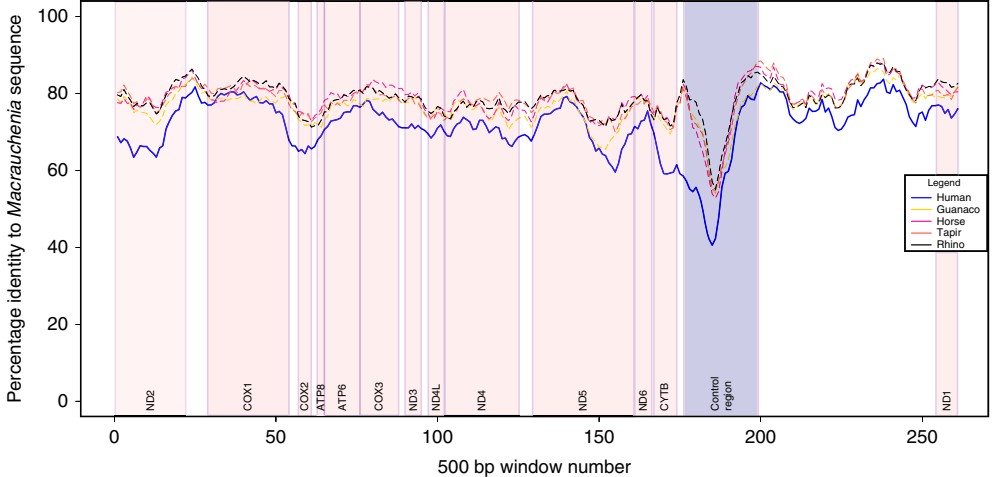

**Figure 2 | Contamination check using pairwise sliding-window comparisons.** Comparisons were undertaken in 500 bp windows with 50 bp overlaps. *X* axis represents the sliding window number. Approximate gene locations within sliding windows are indicated by pink coloured boxes, and the control region is indicated by a blue box. Five sliding window pairwise comparisons are shown: MAC002-human (blue), MAC002-rhino (black), MAC002-guanaco (yellow), MAC002-tapir (orange) and MAC002-horse (red).

read length distributions indicative of aDNA (Supplementary Figs 2 and 3). Retrospective mapping of reads from the other analysed *Toxodon* and *Macrauchenia* samples to our *Macrauchenia* mitochondrial genome sequence produced either very few or no hits. This result suggests that none of the other samples contained detectable quantities of endogenous *Macrauchenia* or *Toxodon* mitochondrial DNA.

**Phylogenetic reconstruction.** We used both Maximum Likelihood and Bayesian inference approaches for phylogenetic tree reconstruction to determine the phylogenetic position of *Macrauchenia*. Both approaches recovered *Macrauchenia* as a sister taxon to the order Perissodactyla, represented by the genera *Hippidion, Equus, Dicerorhinus, Tapirus, Rhinoceros, Ceratotherium, Coelodonta* and *Diceros* (Supplementary Figs 4 and 5). We undertook a molecular dating analysis of the superorder Laurasiatheria with Eulipotyphla specified as outgroup, using four fossil calibrations. The calibrated nodes were: the basal divergence of extant lineages of Laurasiatheria (based on the fossil *Protictis haydenianus*[22]), the basal divergence of extant lineages of Bovidae (based on the fossil *Eotragus noyei*[22]), the basal divergence of extant lineages of Perissodactyla (based on the fossil *Sifrhippus sandrae*[23,24]) and the basal divergence of extant lineages of Carnivora (based on the fossil *Hesperocyon gregarius*[25]). This analysis produced an estimated Panperissodactyla divergence time of 66.15 Ma with a 95% CI of 56.64–77.83 Ma (Fig. 3, Supplementary Table 6). The resulting tree is in good agreement with divergence estimates of other major lineages within Laurasiatheria obtained using larger nuclear DNA data sets[25] (Supplementary Table 7). We additionally investigated the potential variability in divergence time estimates among the individual calibrations used by reanalysing the data set using each calibration independently. The mean estimated age for the basal divergence of Panperissodactyla was broadly similar using the Carnivora, Laurasiatheria or Bovidae calibrations. The mean age obtained using the Perissodactyla calibration was older than that produced using any of the other three calibrations, but the 95% credibility intervals generated using these different calibrations all overlapped except in the case of the Perrissodactyla and Carnivora calibrations (Supplementary Table 6).

## Discussion

We successfully recovered a nearly complete mitochondrial genome for the extinct South American native ungulate, *Macrauchenia*. This allowed us to confidently place *Macrauchenia*, and thus Litopterna, as the sister group of crown Perissodactyla, in agreement with collagen sequences obtained by proteomic analyses[8,15], but with a better-resolved divergence time (~66 Ma, 95% CI of 56.64–77.83 Ma). Our results confirm that Litopterna and Perissodactyla together form (non-exclusively) the unranked taxon Panperissodactyla[8]. Didolodontidae, a 'condylarth' group with North American affinities and thought to be directly ancestral to Litopterna, were present in South America by the earliest part of the Palaeocene but (as far as we currently know) not earlier[26] (Supplementary Note 1). Since there are no accepted litopterns in the Paleogene record of North America, and no crown Perissodactyla in that of South America, stem panperissodactyls most likely dispersed into the latter continent before the divergence of Lipoterna. Thus, fossils, palaeobiography and molecules are mutually concordant in suggesting that early divergences within South American Panperissodactyla probably took place very early in the Cenozoic[8], perhaps immediately subsequent to the K/Pg transition.

*Macrauchenia* samples used for this study came from the southern cone of South America (Fig. 1). Although this portion of the continent is more temperate than the equatorial region, we nonetheless expected DNA preservation to be poor[7]. This is vividly illustrated by the lack of detectable endogenous *Macrauchenia* and *Toxodon* DNA in all but the southernmost *Macrauchenia* sample tested in this study. This result underlines the importance of macro- and microenvironmental factors, with the latter being mostly unknown, affecting DNA survival. A second challenge in analysing the relationships of species such as *Macrauchenia patachonica* comes from the lack of closely related extant relatives which can be used for sequence authentication and detecting potential contamination. This problem arises as the degree to which the genome of an extinct species can be mapped successfully against a corresponding reference genome correlates inversely with phylogenetic distance, which is especially true for the mitochondrial genome with its comparatively rapid substitution rate in vertebrates[27,28].

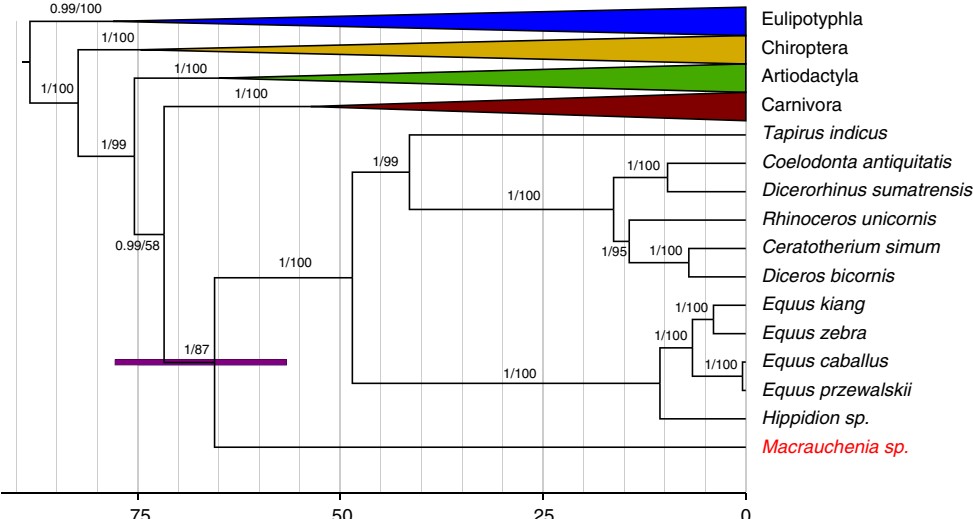

**Figure 3 | Dated mitogenomic phylogenetic tree.** Posterior probabilities and bootstrap values are indicated on the tree branches. The purple node bar represents the 95% CI for the Panperissodactyla clade divergence date based on the combination of all four calibrations used in this study. Scale bar represents time in millions of years. Grey vertical lines represent five million year intervals.

On the other hand, the small size of the mitochondrial genome simplifies the assembly of fossil sequences using *de novo* methods. Such an approach is likely unsuitable for the much larger nuclear genome. As we demonstrate here, iterative mapping strategies permit recovery of nearly complete mitochondrial genomes even from extinct species with only distant living relatives available for comparisons, thereby also permitting resolution of their phylogenetic relationships[29–31]. Caution is, however, required when defining algorithmic parameters or consensus-building steps (Supplementary Table 4) because of the danger of inferring incorrect sequences. The use of a single bait reference sequence may also introduce some biases as final consensus sequences can differ depending on the starting reference bait sequence (Supplementary Table 3). Further, random read mapping of short contaminating DNA sequences is especially likely with aDNA data sets[32] when using less stringent parameters. To control for these and other issues, we adopted a highly conservative approach involving strict minimum coverage and threshold values for consensus construction, a range of mismatch values and the combination of four different bait references, including one phylogenetically more distant sequence. Our method may result in the loss of some correctly sequenced nucleotide sites, but we are confident that all reconstructed positions reported here for *Macrauchenia* are authentic. We suggest that similarly stringent approaches are implemented in future efforts to reconstruct the mitogenomes of extinct organisms without a close living relative, to avoid partially incorrect sequences that may occur with more relaxed approaches.

Although progress in sorting out the molecular palaeontology of Darwin's peculiar mammals is being made, uncertainties remain. The SANU orders Astrapotheria, Pyrotheria and Xenungulata still lack firm placement within Placentalia. Like Notoungulata, Xenungulata has recently been grouped with Afrotheria on morphological grounds[14], implying that this clade is not part of Panperissodactyla. Pyrotheres and xenungulates, never very diverse, had already disappeared by the end of the Palaeogene[33], far beyond the present empirical reach of any molecular method, including collagen proteomics[8]. Prospects may be better for the more diverse astrapotheres, which persisted into the Middle Miocene[34]. However, while we were unable to successfully recover DNA from samples of *Toxodon*, the results from our study underscore the reliability of the collagen results[8,15] and its use in phylogenetic analyses despite the fact that these approaches are methodologically quite different.

## Methods

**Samples.** We carried out genetic analyses on 6 *Macrauchenia* and 11 *Toxodon* subfossils originating from various locations in the southern portion of South America (Fig. 1, Supplementary Table 1). A schematic overview of the methods can be seen in Supplementary Fig. 6.

**DNA preparation.** For all samples, ∼50 mg of bone was ground to powder using a mortar and pestle, and DNA extracted following the protocol described in Dabney *et al.*[18]. Initially, bone powder was rotated at 37 °C in 1 ml extraction buffer (0.45M EDTA, 0.25 mg ml$^{-1}$ proteinase K, pH 8.0). Remaining bone powder was then pelleted by centrifugation at maximum speed (16,000*g*). The supernatant was removed and added to 13 ml of binding buffer (5 M guanidine hydrochloride, 40% isopropanol, 0.05% Tween 20 and 90 mM sodium acetate (pH 5.2)), then passed through a MinElute silica spin column (Qiagen) by centrifugation at 520*g*. The silica membrane was then washed twice by adding 750 μl PE buffer (Qiagen) to the column, centrifuging at 3,300*g* and discarding the flow through. Finally, DNA was eluted from the spin column by adding 25 μl TET buffer followed by an incubation of 5 min and centrifuged at maximum speed for one minute. An additional 50 mg of bone powder from six of these samples, MAC001-004, TOX008 and TOX009 (details in Supplementary Table 8), was pretreated with 1 ml of 0.5% bleach (sodium hypochlorite) for 15 min before DNA extraction, in an attempt to increase endogenous content[35]. All DNA extracts were converted to barcoded Illumina sequencing libraries using a method based on single-stranded DNA specifically developed for highly degraded ancient

samples[5,19]. Extracts initially underwent a uracil excision and DNA cleavage at abasic sites step. 20 μl of DNA extract was added to a solution containing 29 μl water, 8 μl Circligase buffer II (10 ×), 4 μl MnCl$_2$ (50 mM), 0.5 μl Endonuclease VIII (10 U μl$^{-1}$) and 0.5 Afu UDG (2 U μl$^{-1}$). We then incubated this solution at 37 °C for 1 h. Samples then underwent dephosphorylation and denaturation. 1 μl of FastAP (1U) was added to the solution then incubated at 37 °C for 10 min followed by 95 °C for 2 min before being returned to room temperature. The first adaptor was ligated by adding 32 μl PEG-4000 (50%), 1 μl adaptor oligo CL78 (10 μM) and 4 μl Circligase II (100 U μl$^{-1}$) to the 43 μl solution and incubated for 1 h at 60 °C. 20 μl of MyOne C1 beads were pelleted using a magnetic rack, the supernatant was removed and the beads were washed twice with 500 μl of bead binding buffer (1 M NaCl, 10 mM Tris-HCl (pH 8), 0.4 mM EDTA (pH 8), 0.05% Tween 20 and 0.5% SDS). The beads were then resuspended in 250 μl bead binding buffer. Ligation products were then immobilized onto the beads. The adaptor ligation solutions were incubated at 95 °C for 1 min, before being cooled to room temperature and added to the bead buffer solution. The bead suspension was then rotated for 20 min at room temperature. The solution was then pelleted on a magnetic rack, the supernatant was removed and the beads were then washed once with wash buffer A (0.1 M NaCl, 10 mM Tris-HCl, 1 mM EDTA, 0.05% Tween and 0.5% SDS) followed by a wash with wash buffer B (0.1 M NaCl, 10 mM Tris-HCl, 1 mM EDTA and 0.05% Tween). The beads were pelleted and the wash buffer was discarded. Samples then underwent primer annealing and extension. Isothermal amplification buffer of 5 μl (10 ×), 0.5 μl dNTP mix (25 mM each), 1 μl extension primer CL9 (100 μM) and 40.5 μl water was added to the pelleted beads followed by incubation at 65 °C for two minutes before being cooled to room temperature. After incubation, 2 μl Bst 2.0 polymerase (24 U) was added. The mixture was then incubated by increasing the temperature by 1 °C per minute from 15 to 37 °C with a final incubation of 5 min at 37 °C. Beads were then washed once in wash buffer A, once in stringency wash buffer (0.1% SDS and 0.1 × SSC) with an incubation at 45 °C for 4 min and once in wash buffer B. Samples were then blunt end repaired. 10 μl Buffer Tango (10 ×), 2.5 μl Tween 20 (1%), 0.4 μl dNTP (25 mM each), 1 μl T4 polymerase (5U) and 86.1 μl water were added to the pelleted beads followed by a 15 min incubation at 25 °C. Beads were again washed once in wash buffer A, once in stringency wash buffer with an incubation at 45 °C for 4 min and once in wash buffer B. Samples then had the second adaptor ligated. 10 μl T4 DNA ligase buffer (10x), 10 μl PEG-4000 (50%), 2.5 μl Tween 20 (1%), 2 μl double stranded adaptor mixture (100 μM), 2 μl T4 DNA ligase (10U) and 73.5 μl water was added to the beads, followed by a 1 h incubation at room temperature. Beads were again washed once in wash buffer A, once in stringency wash buffer with an incubation at 45 °C for four minutes and once in wash buffer B. Sample bead pellets were then eluted by re-suspension in 25 μl TET buffer followed by an incubation at 95 °C for 1 min and pelleted using a magnetic rack. The supernatant contained the eluted library. Libraries were amplified and indexed by adding 10 μl Accuprime Pfx reaction mix (10 ×), 4 μl P7 indexing primer (10 μM) 4 μl P5 indexing primer (10 μM), 24 μl library, 1 μl Accuprime Pfx polymerase (2.5 U μl$^{-1}$) and 57 μl water followed by a selected number of PCR cycles, involving denaturation for 15 s at 95 °C, annealing for 30 s at 60 °C and primer extension for 1 min at 68 °C. Amplified libraries were cleaned up using a Minelute PCR purification kit following the manufacturer's protocol. Sequencing of library and extraction blanks were included to check for the presence of contamination.

**Test sequencing and analysis.** For each library, we sequenced ∼2–20 million 75 bp PE read pairs on an Illumina Nextseq 500 sequencing platform. Raw Illumina intensity data were demultiplexed and converted to nucleotide sequences using the Illumina software, bcl2fastq. Adaptor sequences were trimmed from the raw reads, reads with lengths of 30 bp or less were discarded using Cutadapt 1.4 (ref. 36). Remaining trimmed reads were then merged using FLASH v1.2.10 (ref. 37). As *Macrauchenia* and *Toxodon* have been previously shown to be related to the order Perissodactyla[8], trimmed and merged reads were mapped to both the horse (GCA_000002305.1) and the rhinoceros (GCA_000283155.1) nuclear genomes using BWA 0.7.8 (ref. 38) to evaluate the presence of endogenous DNA. Duplicate reads and reads of low mapping quality were then removed using SAMtools v0.1.19-44428cd (ref. 39). Endogenous content was estimated by the fraction of merged reads that were successfully mapped to the reference nuclear genomes (Supplementary Table 2).

**Further extractions of MAC002.** As sample MAC002 showed potentially high endogenous DNA content (Supplementary Table 2), we further investigated the potential to increase endogenous DNA recovery, by extracting DNA from this sample a third time using the predigestion method proposed by Damgaard *et al.*[40], but utilizing the extraction buffer and DNA purification method of Dabney *et al.*[18]. Bone powder was mixed with 1 ml of extraction buffer (0.45M EDTA, 0.25 mg ml$^{-1}$ proteinase K, pH 8.0) and incubated for one hour at 37 °C. Samples were then centrifuged for 1 min at maximum speed, the supernatant was removed, added to 13 ml of binding buffer and DNA purified from it following the method described above. The remaining undigested bone powder was then re-extracted by suspension in 1 ml of Dabney extraction buffer followed by an overnight incubation at 37 °C. The bone powder was then subjected to the same extraction procedures as those described above. These extracts were converted into libraries, test sequenced and analysed as described above.

**Deep sequencing of MAC002.** We resequenced all MAC002 libraries produced from the three different extraction methods on an Illumina Nextseq 500. Sample pre-treatment using either bleach or predigestion did not result in libraries with increased endogenous DNA content (Supplementary Table 2). To ensure that only high quality sequences were used for subsequent mitochondrial reconstruction, we processed the raw reads using stringent criteria. First, potential PCR duplicates were removed using fastuniq[41]. Next, we trimmed adaptor sequences and low quality bases from read ends, and discarded any reads <31 bp, using Cutadapt 1.4 (ref. 36). We then merged the trimmed PE reads using FLASH v1.2.10 (ref. 37) with a maximum difference allowed for merging of 0.1. We undertook all further analyses with these trimmed, merged reads (see Supplementary Table 9).

**Mitochondrial reconstruction.** We reconstructed the mitochondrial sequence of MAC002 using MITObim v1.8, a wrapper script for the Mira v4.0.2 (ref. 42) assembler. Direct reconstruction without prior mapping assembly with default parameters was implemented. We initially tested MITObim with the default mismatch value and using four different reference bait mitochondrial sequences, three from the order Perissodactyla and one from the order Artiodactyla. Mira output maf files were then converted to sam files and visualized using Geneious v9.0.5 (ref. 43). Visual inspection revealed that MITObim assemblies generated using default parameters contained very large numbers of spuriously mapped reads or misassemblies, and that these regions were typically associated with either very high or very low coverage.

**MITObim validation.** To better evaluate and optimize the MITObim assembly, we tested the ability of MITObim to reconstruct the correct mitochondrial sequence of a Late Pleistocene cave hyena (*Crocuta crocuta spelaea*), an extinct taxon for which published mitogenome sequences are available that can be used to validate the MITObim assembly. These validation tests used cave hyena shotgun sequencing data with a similar predicted endogenous content and similar number of reads as MAC002. We iteratively mapped these reads to both the dog (*Canis lupus familiaris* GenBank accession: NC002008) and the brown bear (*Ursus arctos*, GenBank HQ685964) mitochondrial genomes, as they are thought to have diverged at a similar phylogenetic time (~50 million years) from *Crocuta* as *Macrauchenia* from the Perissodactyla reference sequences used. We implemented mismatch values ranging from 0 to 12%. Output consensus sequences were then compared to one generated from the same raw reads mapped to a cave hyena mitochondrial genome (Genbank, NC_020670.1) using BWA 0.7.8 (Supplementary Table 4).

**MAC002 mitochondrial genome reconstruction.** We generated assemblies of the MAC002 mitochondrial genome using mismatch values of 0–6% in steps of 1% for each of the four reference sequences described above, resulting in a total of 28 assemblies. Visual examination of these assemblies revealed that both mismatch values and the reference used affected both the regional coverage and regional accumulation of spurious alignments for the assembly. We therefore processed the 28 assemblies into a single consensus sequence, with the aim of maximizing total coverage information, while removing any incongruent sites among assemblies generated using different reference sequences or mismatch values, which may potentially result from analytical bias or contamination. This processing involved three stages of analysis: (1) consolidation of each sequence read assembly into a consensus sequence, (2) consolidation of consensus sequences generated using different references and the same mismatch value into a mismatch value consensus sequence and (3) consolidation of mismatch value consensus sequences into a final consensus sequence.

Initial consensus sequences were generated using strict coverage filters, to ensure that any regions containing incorrect assemblies did not contribute to the final consensus sequence. We estimated the average read depth of coverage for the mitochondrion provided by the MAC002 data to be ~43×. Based on our previous tests using the cave hyena (Supplementary Table 4), we generated consensus sequences by applying a minimum of 34× coverage, representing around 80% of the mean coverage and a maximum coverage of two times the mean read depth (86×). Nucleotide positions within this range were only included in the consensus sequence if a minimum of 95% of reads supported the same nucleotide, or otherwise entered as missing data.

Mismatch value consensus sequences were generated by aligning all consensus sequences (one for each reference used) corresponding to each mismatch value using Mafftv7.123b (ref. 44). A majority rule base call was applied to these alignments to produce a preliminary mismatch value consensus sequence, which was then aligned back to the original consensus sequences and visualized in MEGA6 (ref. 45). Any nucleotide positions that were variable among consensus sequences were scored as missing data (N) in the mismatch value consensus.

The final consensus sequence was generated by aligning all seven mismatch value consensus sequences (one for each mismatch value). The final consensus sequence was then produced as described above, scoring any nucleotide positions that were variable among mismatch value consensus sequences as missing data.

**Final sequence validation.** The online automated mitochondrial annotation programme MITOS[20] was used to evaluate the orientation and positions of tRNAs and protein coding genes within our final consensus sequence. Protein coding

genes were manually identified based on the horse mitochondrial sequence, extracted and translated into their respective amino acid sequences using MEGA6 to check for premature stop codons.

We then calculated pairwise distances for the MAC002 sequence from the four references used for assembly, as well as the human mitochondrial sequence (Genbank accession J01415.2), along a sliding window. Following alignment, sites with missing data were removed manually in MEGA6 and pairwise distances calculated for windows of 500 bp at 50 bp intervals using a custom perl script. Finally, the trimmed and merged MAC002 reads were mapped back to our final consensus sequence using BWA[38] and parsed using samtools[39], allowing investigation of aDNA damage patterns and read length distributions using mapdamage2.0.2–8 (ref. 21; Supplementary Figs 2 and 3).

**Retrospective mapping of other *Macrauchenia* and *Toxodon* samples.** We retrospectively mapped reads from the other analysed samples to our final *Macrauchenia* consensus sequence using BWA 0.7.8 (ref. 38), to better assess the presence of endogenous mitochondrial fragments in these samples.

**Phylogenetic reconstruction.** Our final *Macrauchenia* consensus sequence was aligned with 94 other mitochondrial sequences, including representatives from all major clades of the Laurasiatheria superorder (Supplementary Table 10). All sites containing missing data (N) were removed from the alignment manually, along with the control region, resulting in 12,997 bp of aligned sequence. tRNA and gene positions within this alignment were determined manually for later partitioning analyses.

A maximum likelihood phylogenetic analysis was carried out using Raxml-HPC2 (ref. 46) on XSEDE (on the CIPRES server[47]). The appropriate partitioning scheme for all possible combinations of genes and tRNAs, and appropriate substitution models for each partition (GTR considering all possible combinations of invariant sites and gamma distributed rate heterogeneity parameters) was selected under the Bayesian Information Criterion (BIC) using PartitionFinder[48] (Supplementary Table 11). We did not partition by codon position as the removal of columns with missing data from our alignment would have led to some individual codon partitions being extremely small, confounding the ability to optimize both partitioning scheme and substitution models across all possible combinations of partitions. Furthermore, by including substitution models which accommodate substitution rate heterogeneity among nucleotide positions (+I, +G), our model selection approach is able to accommodate such heterogeneity resulting from codon positioning within individual data partitions. We then carried out five-hundred bootstrap replicates using unlinked GTR+CAT (which approximates the GTR+G model) substitution models for each partition, with a final maximum likelihood tree calculated using GTR+G models. The Eulipotyphla clade was specified as outgroup (comprising *Uropsilus* sp., *Crocidura attenuata*, *Episoriculus caudatus*, *Neomys fodiens*, *Nectogale elegans*, *Uropsilus soricipes*, *Crocidura shantungensis*, *Blarinella quadraticauda*, *Talpa europaea*, *Mogera wogura*, *Galemys pyrenaicus*, *Sorex araneus* and *Anourosorex squamipes*).

Phylogeny and divergence times were then jointly estimated using a Bayesian approach in BEAST 1.8.3 (ref. 49). For the Bayesian approach, we found the appropriate partition scheme and substitutions models through a second run of PartitionFinder[48] (this time including GTR, TrNef, TrN, HKY, K80, HKY or SYM, and again considering all possible combinations of invariant sites and gamma distributed rate heterogeneity parameters; Supplementary Table 12). Time calibration was achieved by applying informative exponentially distributed priors on the ages of four internal nodes of the tree, based on information from the fossil record (Supplementary Table 13). These were: 'Bovidae', incorporating the basal divergence of the bovid clade, which must have occurred before 18 Ma based on the age of the fossil bovid *Eotragus noyei*[22]; 'Carnivora', describing the basal divergence of the carnivore clade, which must have occurred before 37.1 Ma based on the age of the fossil *Hesperocyon gregarius*[25]; 'Perissodactyla', incorporating the basal divergence of the perissodactyl clade, which must have occurred before 47.8 Ma based on the age of the fossil (*Sifrhippus sandrae*[23,24]) and 'Laurasiathera' incorporating the basal divergence of the Laurasiatheria clade, which must have occurred before 62.5 Ma based on the age of the fossil *Protictis haydenianus*[22]. We implemented hard bounds ages of 18.0, 37.1, 47.8 and 62.5 Ma, respectively, and mean ages of 3.357, 6.015, 2.131 and 21.95 Ma, respectively. These parameters gave soft maximum 95% bounds of 28.06, 55.12, 54.18 and 128.3 Ma, respectively. Fossil age distribution values for crown Laurasiatheria, Bovidae and Carnivora were based on those described in Welker *et al.*[8], while the crown Perissodactlya fossil age distribution was based on an early Eocene distribution as fossils are most abundant during this time period[23]. Divergence dates on the tree were estimated using each of the four calibrations in combination and also independently. For each of these analyses, we specified Eulipotyphla as outgroup. An uncorrelated lognormal relaxed clock model was utilized to accommodate variation in substitution rates along individual branches of the tree and a birth-death speciation process[50] was specified as the tree prior. Preliminary runs showed that for some partitions, individual parameters of the substitution model suggested by PartitionFinder[48] failed to converge, indicating over-parameterization. We therefore implemented the simpler HKY+I+G substitution model for these partitions to achieve convergence. The Markov Chain Monte Carlo (MCMC) chain was sampled every 20,000 generations, and ran for a sufficient number of generations to reach

convergence and provide sufficient sampling of the posterior distributions of all parameters (ESS > 200), as determined using the programme Tracerv1.6 (ref. 51). The maximum clade credibility tree was extracted, with node heights scaled to the median of the posterior sample, and visualized using Figtree v1.4.2 (ref. 52).

**Data availability.** The mitochondrial sequence for MAC002 can be found under the accession code KY611394 on Genbank.

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

## Acknowledgements

We thank members of the following institutions for permission to sample specimens in their care: in Argentina, the Museo Argentino de Ciencias Naturales 'Bernardino Rivadavia' (Buenos Aires), Museo de La Plata, Museo Municipal de Ciencias Naturales 'Lorenzo Scaglia' (Mar del Plata) and Museo Paleontológico de San Pedro 'Fray Manuel de Torres' (San Pedro); in Chile, the Departamento de Antropología (Universidad de Chile) and the Facultad de Patrimonio Cultural y Educación (Universidad SEK Chile); in Uruguay, Museo Nacional de Historia Natural (Montevideo); and in France, Muséum national d'Histoire naturelle (Paris). All fossil samples were collected before 2010. We also thank Rheon Slade for assisting in the modifications of some figures. This work was partly supported by National Science Foundation DEB 1547414 (R.D.E.M.). This work was also supported by the European Research Council (consolidator grant GeneFlow # 310763 to M.H.). The NVIDIA TITAN X GPU used for BEAST analyses was kindly donated by the NVIDIA Corporation.

## Author contributions

R.D.E.M. and M.H. conceived the project; M.W. and S.B. performed lab work; M.W., S.B., A.B., S.H. and J.L.A.P. performed DNA analyses and interpretation of results; M.W. and A.B. conducted the phylogenetic analyses and constructed trees; A.K., A.M.F., M.B., J.N.G., M.A.R., P.L.M., M.T., F.S., A.R., W.J., G.B., C.d.M., F.M. and J.L.A. assisted with locating and sampling specimens or provided logistical help; A.K., A.M.F., M.B., J.N.G.,

M.A.R., P.L.M. and R.D.E.M. provided the palaeontological and systematic framework for this study and wrote the palaeontological portion of the supplementary file. Final editing and manuscript preparation was coordinated by M.W., A.B., R.D.E.M. and M.H. All contributing authors read and agreed to the final manuscript.

## Additional information

**Competing interests:** The authors declare no competing financial interests.

