## [Peer Review File · Nature Communications]

Reviewers' comments:

Reviewer #1 (Remarks to the Author):

The affinities of native South American ungulates have been problematic since their discovery, and has become even more enigmatic as more detailed anatomical evidence has accrued through the past century. The fact that each of these five or so mammalian orders is entirely extinct has added to the allure of the problem, and also has served to limit the nature of relevant data to anatomical features that can be observed in fossils (and often quite incomplete fossils; many of the early taxa are known by little more than teeth). In this context, the extraction and reconstruction of the mitochondrial genome belonging to a litoptern (the late-surviving genus *Macrauchenia*) is hugely important. The authors support the hypothesis that *Macrauchenia* (and hence the order Litopterna) belongs to Laurasiatheria and is most closely related to perissodactyls—an idea that was forwarded nearly 130 years ago by Florentino Ameghino and that has subsequently been dismissed by pretty much everyone. The results and their data source are so novel that this report constitutes headline news: in short, this paper marks a watershed in phylogenetic reconstruction for major extinct groups. I urge Nature to publish this work.

I am largely ignorant of the methods and analytic techniques. But I have combed through as best I could, and as I said, I think the authors have approached the study with proper caution and have done what they can to make their results robust.

I will admit to one self-serving comment: I think one or more of my studies should be mentioned, however briefly, in the section in the SI dealing with previous work on the affinities of Litopterna, partly because I worked extensively on relationships within the order, and partly because I addressed their possible relationships to non-SA groups:

Cifelli, R.L. 1983. Eutherian tarsals from the late Paleocene of Brazil. *American Museum Novitates* 2761: 1-31.

Cifelli, R.L. 1983. The origin and affinities of the South American Condylarthra and early Tertiary Litopterna (Mammalia). *American Museum Novitates* 2772: 1-49.

Cifelli, R.L. 1993. Paleobiology of *Megadolodus*, an unusual ungulate from the Miocene of Colombia. *Journal of Vertebrate Paleontology* 13, supplement: 30A.

Cifelli, R.L. 1993. The phylogeny of the native South American ungulates. In: F.S. Szalay, M.J. Novacek, and M.C. McKenna (eds.), *Mammal Phylogeny: Placentals*, 195-216. Springer-Verlag, New York.

Cifelli, R.L. and Soria, M.F. 1983. Systematics of the Adiantidae (Litopterna, Mammalia). *American Museum Novitates* 2771: 1-25.

Reviewer #2 (Remarks to the Author):

In this manuscript Westbury et al. present a near-whole mitochondrial genome of an individual from an extinct ungulate family, whose position in the phylogeny of mammals has

been debated. I don't think this ends the discussion regarding the other groups of South American native ungulates, but it is strong evidence for this group. The methods and analyses are well done, and so, in addition to being important for our understanding of mammal evolution, this paper will assist in the ancient genomic analysis of other extinct groups without close relatives. I enjoyed reading the manuscript, and my few comments are minor.

The quality of the figures in the Supplementary materials is too poor to read the labels.

Minor edits:

Line 114- extra period in the middle of the sentence

Line 328- period missing at end of sentence

Reviewer #3 (Remarks to the Author):

In this paper, the authors present a nearly complete mitochondrial genome of the South American "native ungulate" (SANU) *Macrauchenia*, for which previously the only molecular data available was collagen amino acid sequences. The issue of where SANUs "fit" within the placental tree has been controversial, and it is nice to see that the mitochondrial data agrees with the collagen data in supporting a relationship with perissodactyls. The authors also employ (as far as I can judge) a novel approach for identifying authentic ancient DNA of *Macrauchenia* and for reconstructing the mitochondrial genome in the absence of close relatives to act as a reference. Overall, I consider the paper both interesting and important, and it is also very well written - as such, it is definitely suitable for publication in *Nature Communications*.

I am not an ancient DNA specialist, so I cannot really give any sensible comments on the methods used in sequencing and reconstructing the genome - I hope at least one other reviewer is an aDNA person! I do, however, know something about phylogenetic analysis and mammal systematics, so I will focus my comments on these. Some of the methods/assumptions made in the phylogenetic analyses are a bit questionable - I suspect that they won't have that big an impact on the resultant phylogeny, but I can't be sure. However, my comments on fossil calibrations definitely need to be addressed.

My main comments are as follows:

1. Why weren't the 12S and 16S rRNA genes sequenced?
2. In doing your PartitionFinder analysis, you appear to have initially partitioned the alignment by gene, but not by codon position. It is clear the mitochondrial sequences in particular show major differences in base composition between codon positions, so grouping them all together as, for example, "cytb", is problematic. I think it is much more important to separate out the 3rd codon positions in the initial partitioning scheme - I strongly suspect that PartitionFinder will not end up grouping the 1st, 2nd and 3rd positions of a particular

gene together.

3. I'm a bit confused how the partitioning scheme preferred by PartitionFinder relates to the RAxML analyses. RAxML only implements versions of GTR (GTR, GTR+G, GTR+I, GTR+I+G - Stamatakis only really recommends use of GTR+G, if you read the RAxML manual), so really you should repeat your PartitionFinder analysis specifying that only versions of GTR are tested, and then use that output for your RAxML analyses

4. It is inappropriate to use a Yule prior for this dataset - a Yule prior is a pure birth model and can result in misleading results, e.g. Condamine, F. L., Nagalingum, N. S., Marshall, C. R., and Morlon, H. (2015). Origin and diversification of living cycads: a cautionary tale on the impact of the branching process prior in Bayesian molecular dating. *BMC Evolutionary Biology* 15: 65. BEAST also implements a birth-death model, which I would recommend using here.

5. I have concerns about some of the divergence dates that result here - at least some of them are incongruent with the fossil record in being much *younger* than the oldest fossils referable to descendant lineages. For example, your date for Perissodactyla (Hippomorpha - Ceratomorpha split) has a mean estimate of 31.74 MYA, which is markedly younger than the oldest fossils referable to Hippomorpha and Ceratomorpha, which are ~52 MYA. I suspect that this is because you have used calibrations within Laurasiatheria that belong to relatively large-bodied lineages - see Phillips, M. J. (2016). Geomolecular dating and the origin of placental mammals. *Systematic Biology* 65: 546-557 for an interesting discussion of this issue. Choosing and implementing fossil calibrations is always going to be something of a subjective exercise, but the timing of the Macrauchenia-Perissodactyla split is critically important for, for example, relating this to the dispersal of placentals into South America from North America, and interpreting the putative fossil litopterns from Punta Peligro and Itaborai. I strongly recommend reconsidering your choice of calibrations (or at least adding a few smaller bodied lineages, e.g. for bats - see the Phillips paper already mentioned, plus <http://palaeo-electronica.org/content/fc-5>), together with the way you have partitioned your data and your choice of a Yule prior. I suspect that addressing these issues will push your estimated dates back a bit.

6. The discussion of litoptern systematics in the supplementary info is great, and it's a shame it's buried away here! It would be nice to see a version of this (perhaps expanded) published as a standalone paper at some point.

See also my minor, nitpicky comments indicated on the pdf itself.

Robin M.D. Beck
r.m.d.beck@salford.ac.uk

Reviewers' responses for the manuscript : Westbury et al., A mitogenomic timetree for Darwin's enigmatic "transitional" South American mammal, *Macrauchenia patachonica*

Reviewer #1 (Remarks to the Author):

The affinities of native South American ungulates have been problematic since their discovery, and has become even more enigmatic as more detailed anatomical evidence has accrued through the past century. The fact that each of these five or so mammalian orders is entirely extinct has added to the allure of the problem, and also has served to limit the nature of relevant data to anatomical features that can be observed in fossils (and often quite incomplete fossils; many of the early taxa are known by little more than teeth). In this context, the extraction and reconstruction of the mitochondrial genome belonging to a litoptern (the late-surviving genus *Macrauchenia*) is hugely important. The authors support the hypothesis that *Macrauchenia* (and hence the order Litopterna) belongs to Laurasiatheria and is most closely related to perissodactyls—an idea that was forwarded nearly 130 years ago by Florentino Ameghino and that has subsequently been dismissed by pretty much everyone.

The results and their data source are so novel that this report constitutes headline news: in short, this paper marks a watershed in phylogenetic reconstruction for major extinct groups. I urge Nature to publish this work.

I am largely ignorant of the methods and analytic techniques. But I have combed through as best I could, and as I said, I think the authors have approached the study with proper caution and have done what they can to make their results robust.

I will admit to one self-serving comment: I think one or more of my studies should be mentioned, however briefly, in the section in the SI dealing with previous work on the affinities of Litopterna, partly because I worked extensively on relationships within the order, and partly because I addressed their possible relationships to non-SA groups:

Cifelli, R.L. 1983. Eutherian tarsals from the late Paleocene of Brazil. *American Museum Novitates* 2761: 1-31.

Cifelli, R.L. 1983. The origin and affinities of the South American Condylarthra and early Tertiary Litopterna (Mammalia). *American Museum Novitates* 2772: 1-49.

Cifelli, R.L. 1993. Paleobiology of *Megadolodus*, an unusual ungulate from the Miocene of Colombia. *Journal of Vertebrate Paleontology* 13, supplement: 30A.

Cifelli, R.L. 1993. The phylogeny of the native South American ungulates. In: F.S. Szalay, M.J. Novacek, and M.C. McKenna (eds.), *Mammal Phylogeny: Placentals*, 195-216. Springer-Verlag, New York.

Cifelli, R.L. and Soria, M.F. 1983. Systematics of the Adianthidae (Litopterna, Mammalia). *American Museum Novitates* 2771: 1-25.

Response: We have added the appropriate citations to the discussion section of the main text and supplementary information.

Reviewer #2 (Remarks to the Author):

In this manuscript Westbury et al. present a near-whole mitochondrial genome of an individual from an extinct ungulate family, whose position in the phylogeny of mammals has been debated. I don't think this ends the discussion regarding the other groups of South American native ungulates, but it is strong evidence for this group. The methods and analyses are well done, and so, in addition to being important for our understanding of mammal evolution, this paper will assist in the ancient genomic analysis of other extinct groups without close relatives. I enjoyed reading the manuscript, and my few comments are minor.

The quality of the figures in the Supplementary materials is too poor to read the labels.

Minor edits:

Line 114- extra period in the middle of the sentence

Line 328- period missing at end of sentence

Response: We have adjusted figure quality and made the suggested corrections.

Reviewer #3 (Remarks to the Author):

In this paper, the authors present a nearly complete mitochondrial genome of the South American "native ungulate" (SANU) *Macrauchenia*, for which previously the only molecular data available was collagen amino acid sequences. The issue of where SANUs "fit" within the placental tree has been controversial, and it is nice to see that the mitochondrial data agrees with the collagen data in supporting a relationship with perissodactyls. The authors also employ (as far as I can judge) a novel approach for identifying authentic ancient DNA of *Macrauchenia* and for reconstructing the mitochondrial genome in the absence of close relatives to act as a reference. Overall, I consider the paper both interesting and important, and it is also very well written - as such, it is definitely suitable for publication in *Nature Communications*.

I am not an ancient DNA specialist, so I cannot really give any sensible comments on the methods used in sequencing and reconstructing the genome - I hope at least one other reviewer is an aDNA person! I do, however, know something about phylogenetic analysis and mammal systematics, so I will focus my comments on these. Some of the methods/assumptions made in the phylogenetic analyses are a bit questionable - I suspect that they won't have that big an impact on the resultant phylogeny, but I can't be sure. However, my comments on fossil calibrations definitely need to be addressed.

My main comments are as follows:

1. Why weren't the 12S and 16S rRNA genes sequenced?

Response: These genes were sequenced and used for analysis. Failure to report this in the main text was a mistake on our part, and the appropriate corrections have been made to the Results section: “*Macrauchenia* mitochondrial sequence validation”.

2. In doing your PartitionFinder analysis, you appear to have initially partitioned the alignment by gene, but not by codon position. It is clear the mitochondrial sequences in particular show major differences in base composition between codon positions, so grouping them all together as, for example, "cytb", is problematic. I think it is much more important to separate out the 3rd codon positions in the initial partitioning scheme - I strongly suspect that PartitionFinder will not end up grouping the 1st, 2nd and 3rd positions of a particular gene together.

Response: We decided against partitioning by codon position due to large amounts of missing data within our *Macrauchenia* sequence, which necessitated the removal of the respective columns from the sequence alignment. As a result, some individual codon partitions would have been extremely small, confounding the ability to optimise both partitioning scheme and substitution models across all possible combinations of partitions. Furthermore, by testing substitution models which accommodate substitution rate heterogeneity among nucleotide positions (+I, +G), our model selection approach is able to accommodate such heterogeneity resulting from codon positioning within individual data partitions. We have added text to the methods section further describing our model testing approach and the limitation resulting from missing data removal.

3. I'm a bit confused how the partitioning scheme preferred by PartitionFinder relates to the RAxML analyses. RAxML only implements versions of GTR (GTR, GTR+G, GTR+I, GTR+I+G - Stamatakis only really recommends use of GTR+G, if you read the RAxML manual), so really you should repeat your PartitionFinder analysis specifying that only versions of GTR are tested, and then use that output for your RAxML analyses

Response: We have rerun PartitionFinder using only the GTR family of models available in Raxml and have adjusted the manuscript accordingly.

4. It is inappropriate to use a Yule prior for this dataset - a Yule prior is a pure birth model and can result in misleading results, e.g. Condamine, F. L., Nagalingum, N. S., Marshall, C. R., and Morlon, H. (2015). Origin and diversification of living cycads: a cautionary tale on the impact of the branching process prior in Bayesian molecular dating. BMC Evolutionary Biology 15: 65. BEAST also implements a birth-death model, which I would recommend using here.

Response: We have rerun analyses using the birth-death prior instead of the yule prior and adjusted the manuscript accordingly.

5. I have concerns about some of the divergence dates that result here - at least some of them are incongruent with the fossil record in being much *younger* than the oldest fossils referable to descendant lineages. For example, your date for Perissodactyla (Hippomorpha-Ceratomorpha split) has a mean estimate of 31.74 MYA, which is markedly younger than the oldest fossils referable to Hippomorpha and Ceratomorpha, which are ~52 MYA. I suspect that this is because you have used calibrations within Laurasiatheria that belong to relatively large-bodied lineages - see Phillips, M. J. (2016). Geomolecular dating and the origin of placental mammals. *Systematic Biology* 65: 546-557 for an interesting discussion of this issue. Choosing and implementing fossil calibrations is always going to be something of a subjective exercise, but the timing of the Macrauchenia-Perissodactyla split is critically important for, for example, relating this to the dispersal of placentals into South America from North America, and interpreting the putative fossil litopterns from Punta Peligro and Itaborai. I strongly recommend reconsidering your choice of calibrations (or at least adding a few smaller bodied lineages, e.g. for bats - see the Phillips paper already mentioned, plus <http://palaeo-electronica.org/content/fc-5>), together with the way you have partitioned your data and your choice of a Yule prior. I suspect that addressing these issues will push your estimated dates back a bit.

Response: The reviewer raises some valid points which we have carefully considered and revised our analytical methods accordingly. In light of the Phillips' article, we have rethought the problem of soft maxima and favor the approach that looks at the density of the fossil record for a clade around its supposed initiation. For Perissodactyla, there is overwhelming evidence that equoids were already in existence in multiple locations (North America, southern Asia, India) at the start of the Eocene. As calibrant we selected *Sifrhippus sandrae*, from basal Wasatchian (Wa 0) western North America, and justified its selection in the text. As well as the addition of this taxon as a fourth calibration, we reanalysed the data using a birth-death speciation prior and exponential prior distributions on calibrated nodes as suggested by the reviewer. We also enforced Eulipotyphla as outgroup. As predicted by the reviewer, these changes have pushed our divergence dates back and they are now more convergent with the fossil records. With respect to the bat calibration suggested by the reviewer we attempted to implement this calibration but the Bayesian analysis failed to converge even after 200,000,000 iterations. Based on this result, which mirrors the difficulty other workers have had in placing Chiroptera within Laurasiatheria, we decided to leave the bat calibration out.

With respect to the reviewers comment on our selection of calibrations and the potentially confounding effects of large body size, truly large body sizes (> 500 kg) as estimated in a wide range of investigations become a general feature of euungulate evolution only with the rise of grasslands from the mid-Miocene onward. Thus for at least half of the time clade Perissodactyla has existed according to the divergence age used here, members would not have been particularly large-bodied, and this is borne out by the fossil record. This is also true for artiodactyls; with the exception of whales, truly large body sizes did not become common in this group until the explosive diversification of Bovidae, again in the Miocene. As to inclusion of

taxa with smaller body sizes as a balance, crown Laurasiatheria contains a wide range of species with small as well as large body sizes.

Globally warm temperatures like those experienced in the PETM are inversely correlated with larger body sizes in the same lineages (Secord et al., 2012, cited in paper), which shows that diet, or longevity, or any other single factor is unlikely to have a persistent effect over long periods on rates. In short, body size is as labile a characteristic as anything else in the phenotype, and defining a long-term trajectory for any major clade is a very fraught problem that cannot be handled by simpling juggling the list of calibrants.

6. The discussion of litoptern systematics in the supplementary info is great, and it's a shame it's buried away here! It would be nice to see a version of this (perhaps expanded) published as a standalone paper at some point.

Response: We thank the reviewer for this comment. There are such discussions already in the Spanish literature, but the reviewer's interest suggests that an overview in English of current analyses and concepts in SANU systematics would be useful.

REVIEWERS' COMMENTS:

Reviewer #3 (Remarks to the Author):

I am pleased to see that the authors have taken the time to fully respond to my comments. As such, I am happy for the paper to be published, once the following very minor points are addressed.

1. The authors provide justification for why they did not initially partition the protein-coding genes by codon position prior to PartitionFinder analysis, namely that it would result in very small partitions for which it might not be possible to identify a suitable partitioning scheme. An alternative approach would have been to treat all of the 12 "H-stand" protein-coding genes (i.e. all except MTND6) as a "supergene" that could be then be partitioned by codon position, i.e. 12mt_1st, 12mt_2nd and 12mt_3rd. The authors mention that the +I and +G parameters will model rate heterogeneity, which is fair enough, but grouping all the codon positions together means that the base composition parameters will be estimated as some kind of average across all codon positions, when we know that base composition of the third codon position differs markedly from that of the first two. However, I suspect that doing this will not have a major impact on the results, and don't expect the authors to redo their analyses once again.

2. It is comforting that revising the choice of calibrations results in dates that are in much better agreement with the fossil record. Perissodactyls apparently never reached South America, and the point estimate for the perissodactyl-SANU split is ~66MA, which is within the probable window of dispersal by SANUs to South America (i.e. after the "Allenian/Alamitian" but before Punta Peligro, so is it possible that the date of the perissodactyl-SANU split approximates the time of dispersal into South America?

3. The authors list the minimum and mean values for the offset-exponential calibrations they used. However, the mean values were presumably selected to generate an appropriate "soft" (95%) maximum bound - what was the maximum bound for each of the four calibrations, and what is the justification for each?

I am happy for my identity to be made known to the authors.

Dr. Robin Beck
Lecturer in Biology | School of Environment & Life Sciences
Room G48, Peel Building, University of Salford, Salford M5 4WT, UK
t: +44 (0)161-295-4994
R.M.D.Beck@salford.ac.uk |

REVIEWERS' COMMENTS:

Reviewer #3 (Remarks to the Author):

I am pleased to see that the authors have taken the time to fully respond to my comments. As such, I am happy for the paper to be published, once the following very minor points are addressed.

1. The authors provide justification for why they did not initially partition the protein-coding genes by codon position prior to PartitionFinder analysis, namely that it would result in very small partitions for which it might not be possible to identify a suitable partitioning scheme. An alternative approach would have been to treat all of the 12 "H-stand" protein-coding genes (i.e. all except MTND6) as a "supergene" that could be then be partitioned by codon position, i.e. 12mt_1st, 12mt_2nd and 12mt_3rd. The authors mention that the +I and +G parameters will model rate heterogeneity, which is fair enough, but grouping all the codon positions together means that the base composition parameters will be estimated as some kind of average across all codon positions, when we know that base composition of the third codon position differs markedly from that of the first two. However, I suspect that doing this will not have a major impact on the results, and don't expect the authors to redo their analyses once again.

Response: The reviewer suggests an alternative approach (concatenation of codon positions across genes into a "supergene") to overcome the problem of small individual data partitions if the alignment were to be separated by both gene and codon position. However, this approach fails to accommodate variability in substitution patterns and rates between genes, which may occur due to differing evolutionary constraints on protein form and function. Clearly, neither approach (individual genes vs. "supergene" codon positions) is perfect and the ideal solution is a more complete alignment, which is beyond the reach of this study. Determining which of these two approaches performs better is likely to be highly dataset specific, and difficult to test, and we agree wholeheartedly with the reviewers assessment that the overall impact on the results is likely to be marginal. We therefore follow the course of action suggested by the reviewer and have conducted no further analysis.

2. It is comforting that revising the choice of calibrations results in dates that are in much better agreement with the fossil record. Perissodactyls apparently never reached South America, and the point estimate for the perissodactyl-SANU split is ~66MA, which is within the probable window of dispersal by SANUs to South America (i.e. after the "Allenian/Alamitian" but before Punta Peligro, so is it possible that the date of the perissodactyl-SANU split approximates the time of dispersal into South America?

Response: We agree that it is entirely possible and have added a paragraph addressing this to the discussion section.

3. The authors list the minimum and mean values for the offset-exponential calibrations they used. However, the mean values were presumably selected to generate an appropriate "soft" (95%) maximum bound - what was the maximum bound for each of the four calibrations, and what is the justification for each?

Response. We have added the 95% soft maximum bound and a sentence explaining the calibration distribution decisions to the methods sections of the manuscript.

I am happy for my identity to be made known to the authors.

Dr. Robin Beck

Lecturer in Biology | School of Environment & Life Sciences

Room G48, Peel Building, University of Salford, Salford M5 4WT, UK

t: +44 (0)161-295-4994

R.M.D.Beck@salford.ac.uk |